# Mitigating Forgetting in Continual Learning with Selective Gradient Projection

## Abstract

As neural networks are increasingly deployed in dynamic environments, they face the challenge of catastrophic forgetting, the tendency to overwrite previously learned knowledge when adapting to new tasks, resulting in severe performance degradation on earlier tasks. We propose Selective Forgetting-Aware Optimization (SFAO), a dynamic method that regulates gradient directions via cosine similarity and per-layer gating, enabling controlled forgetting while balancing plasticity and stability. SFAO selectively projects, accepts, or discards updates using a tunable mechanism with efficient Monte Carlo approximation. Experiments on standard continual learning benchmarks show that SFAO achieves competitive accuracy with markedly lower memory cost, a 90% reduction, and improved forgetting on MNIST datasets, making it suitable for resource-constrained scenarios.

## 1 Introduction

Deep neural networks exhibit remarkable proficiency under static environments but degrade significantly in non-stationary learning environments, where the input-output distribution evolves over time (Parisi et al., 2019). In Continual Learning (CL), where models must learn a sequence of tasks without revisiting previous data, this degradation manifests as catastrophic forgetting (Goodfellow et al., 2013). The root cause lies in gradient-induced interference, whereby updates for new tasks disrupt previously consolidated knowledge, causing subspace collapse in the parameter space and destabilizing learned representations (Lopez-Paz & Ranzato, 2022).

This challenge is particularly acute in safety critical domains such as autonomous driving, medical diagnostics, and cybersecurity, where models must adapt to emerging patterns such as evolving traffic scenarios, novel disease classes, or new malware signatures without compromising prior expertise (Hamedi et al., 2025). Failure to maintain stability in such contexts leads to diminished reliability, costly retraining, and large computational overhead (Armstrong & Clifton, 2022; Lesort, 2020). Consequently, mitigating forgetting while preserving adaptability remains a foundational objective in CL research.

We introduce SFAO, an approach that selectively regulates gradient updates. On each layer, SFAO either accepts, projects, or discards a step based on the cosine alignment with previously stored directions. This provides a lightweight and tunable mechanism, which can be used for controlling updates without requiring a large memory buffers or fixed regularization.

### 1.1 Contributions

1. A simple per-layer gating rule that accepts, projects, or discards updates based on cosine similarity, offering a controllable way to manage gradient updates.

2. A gradient filtering mechanism that discards conflicting or uninformative updates, enhancing knowledge retention and improving generalization across sequential tasks.

3. A conceptually simple optimizer that achieves strong memory-forgetting trade-offs without relying on state-of-the-art accuracy.

## 2 PRELIMINARIES

### 2.1 CONTINUAL LEARNING

In continual learning (CL), a model is trained on a sequence of $T$ tasks

$$\mathcal{D}_1, \mathcal{D}_2, \ldots, \mathcal{D}_T,$$

where each task $\mathcal{D}_t = \{(x_i^{(t)}, y_i^{(t)})\}_{i=1}^{n_t}$ is sampled from a distribution $\mathcal{P}_t(x, y)$. Unlike classical i.i.d. training, the distributions $\{\mathcal{P}_t\}$ are non-stationary and past data $\mathcal{D}_1, \ldots, \mathcal{D}_{t-1}$ is typically inaccessible when training on $\mathcal{D}_t$.

The model parameters $\theta$ are updated using stochastic gradient-based optimization techniques

$$g_t = \nabla_\theta \mathcal{L}_t(\theta),$$

where $\mathcal{L}_t$ is the loss for task $t$. A central challenge is *catastrophic forgetting*: learning new tasks degrades performance on earlier tasks. Formally, the forgetting on task $i$ after all $T$ tasks is

$$F_i = \max_{t \leq T} a_{i,t} - a_{i,T},$$

where $a_{i,t}$ denotes accuracy on task $i$ after training task $t$. To better quantify the ability for a model to remain robust to new tasks, we use *average forgetting*, defined as $F = \frac{1}{T-1} \sum_{i=1}^{T-1} F_i$. Additional measures include *Average Accuracy* (mean accuracy across all tasks at the end of training), *Backward Transfer* (BWT), and the *Plasticity–Stability Measure* (PSM), which together capture the tradeoff between learning new knowledge and retaining old knowledge.

### 2.2 GRADIENT INTERFERENCE: A GEOMETRIC AND FIRST-ORDER VIEW

Let $\{\mathcal{D}_i\}_{i=1}^{t-1}$ denote previously learned tasks with losses $\{\mathcal{L}_i\}$ and let $\mathcal{L}_t$ be the current task. Write $g_i(\theta) = \nabla_\theta \mathcal{L}_i(\theta)$ and $g_t(\theta) = \nabla_\theta \mathcal{L}_t(\theta)$. For a small step $\theta^+ = \theta - \eta u$ (learning rate $\eta > 0$ and update direction $u$), a first-order Taylor expansion gives the instantaneous change on a past task $i$:

$$\Delta \mathcal{L}_i \triangleq \mathcal{L}_i(\theta^+) - \mathcal{L}_i(\theta) = -\eta\, g_i^\top u + O(\eta^2). \tag{1}$$

**Interference** on task $i$ occurs when $g_i^\top u < 0$ (loss increases); **synergy** occurs when $g_i^\top u > 0$ (loss decreases). Define the *interference risk* of an update $u$ against a set $\mathcal{G} \subset \mathbb{R}^d$ of stored directions by

$$\mathcal{R}(u; \mathcal{G}) = \max_{g \in \mathcal{G}} \left( -g^\top u \right)_+, \qquad (x)_+ := \max\{x, 0\}. \tag{2}$$

Minimizing risk, $\mathcal{R}$, encourages $g^\top u \geq 0$ for all $g \in \mathcal{G}$ in the small-step regime, which by equation 1 eliminates first-order forgetting on the represented directions.

Let $\mathcal{S} = \mathrm{span}(\mathcal{G})$ and $P_\mathcal{S}$ be the *orthogonal* projector onto $\mathcal{S}$. Consider the feasibility cone

$$\mathcal{C} = \{u \in \mathbb{R}^d : g^\top u \geq 0 \;\; \forall g \in \mathcal{G}\}. \tag{3}$$

An interference-safe step can be posed as the inequality-constrained Euclidean projection

$$\min_{u \in \mathbb{R}^d} \tfrac{1}{2} \|u - g_t\|_2^2 \quad \text{s.t.} \quad g^\top u \geq 0 \quad \forall\, g \in \mathcal{G}. \tag{4}$$

Problem equation 4 projects $g_t$ onto the polyhedral cone $\mathcal{C}$ and its solution *need not* be orthogonal to $\mathcal{S}$.

A stricter surrogate is the equality-constrained projection

$$\min_{u \in \mathbb{R}^d} \tfrac{1}{2} \|u - g_t\|_2^2 \quad \text{s.t.} \quad g^\top u = 0 \quad \forall\, g \in \mathcal{G}, \tag{5}$$

which enforces $u \in \mathcal{S}^\perp$ and whose solution is obtained by solving the Lagrangian (Appendix C):

$$u^\star = (I - P_\mathcal{S})\, g_t. \tag{6}$$

**Proposition 2.1 (First-order safety for represented tasks)** *If $u = (I - P_\mathcal{S})\, g_t$, then $g^\top u = 0$ for all $g \in \mathcal{S}$, and thus for any past task $i$ whose gradient $g_i \in \mathcal{S}$ we have $\Delta \mathcal{L}_i = O(\eta^2)$. Hence orthogonal projection removes first-order forgetting on tasks whose gradients are represented in $\mathcal{S}$.*

*Proof.* For $g \in \mathcal{S}$ we have $P_\mathcal{S} g = g$, so $g^\top (I - P_\mathcal{S}) g_t = (P_\mathcal{S} g)^\top g_t - g^\top g_t = 0$. Plug into equation 1.

## 2.3 ORTHOGONAL GRADIENT DESCENT (OGD)

Orthogonal Gradient Descent (OGD) (Farajtabar et al., 2019) is a geometry-based continual learning method which addresses gradient interference by constraining updates to directions orthogonal to past gradients. Let $\mathcal{S} = \text{span}\{g_1, \ldots, g_N\}$ be the subspace of stored gradients. OGD projects a new gradient $g_t$ onto the orthogonal complement of $\mathcal{S}$:

$$g_t^\perp = \text{Proj}_{\mathcal{S}^\perp}(g_t) = g_t - \sum_{i=1}^{N} \frac{g_t^\top g_i}{\|g_i\|^2} g_i.$$

This guarantees that the update does not interfere with previously learned directions, thereby preserving earlier task performance. OGD's geometric clarity makes it an appealing baseline, but it is computationally costly: storing all or a large subset of past gradients requires $O(Nd)$ memory (for $d$-dimensional gradients), and each update involves $O(Nd)$ dot products. Subsequent works have sought to approximate this projection using low-rank subspaces or memory buffers to improve scalability.

## 3 SELECTIVE FORGETTING-AWARE OPTIMIZER

### 3.1 SIMILARITY-GATED UPDATE RULE (SFAO)

Let $\theta_t \in \mathbb{R}^d$ denote the parameters at step $t$ and $g_t = \nabla_\theta \mathcal{L}_t(\theta_t)$ the mini-batch gradient. We maintain a buffer of past gradients with span $\mathcal{S} = \text{span}\{g_1, \ldots, g_N\}$ and orthogonal projector $P_{\mathcal{S}}$.

Let $Q \in \mathbb{R}^{d \times r}$ be an *orthonormal* basis for $\mathcal{S}$ (e.g., incremental Gram–Schmidt or compact SVD), so $P_{\mathcal{S}} = QQ^\top$.

Given a Monte Carlo subset $\mathcal{C} \subseteq \{1, \ldots, N\}$ of size $k \ll N$, define the sampled maximum cosine alignment

$$s_t = \max_{i \in \mathcal{C}} \frac{g_t^\top g_i}{\|g_t\| \|g_i\|}. \tag{7}$$

Because $\mathcal{C} \subseteq \{1, \ldots, N\}$, $s_t$ is a deterministic *lower bound* on the true maximum alignment over the buffer.

Choose thresholds $\lambda_{\text{proj}} \leq \lambda_{\text{accept}}$ in $[-1, 1]$ and, if one wishes to accept only synergistic updates, set $\lambda_{\text{accept}} \geq 0$. Then the SFAO *gated direction* $u_t$ is

$$u_t = \begin{cases} g_t, & \text{if } s_t > \lambda_{\text{accept}} \quad (\text{accept}) \\ (I - P_{\mathcal{S}}) g_t = (I - QQ^\top)g_t, & \text{if } \lambda_{\text{proj}} < s_t \leq \lambda_{\text{accept}} \quad (\text{project}) \\ 0, & \text{if } s_t \leq \lambda_{\text{proj}} \quad (\text{discard}) , \end{cases} \tag{8}$$

followed by the SGD-style parameter update

$$\boxed{\theta_{t+1} = \theta_t - \eta\, u_t}. \tag{9}$$

**Recovering special cases (corrected).**

- **SGD**: empty buffer or $\lambda_{\text{accept}} = -1 \Rightarrow u_t = g_t$.
- **Always-project (OGD behavior)**: set $\lambda_{\text{proj}} = -1$, $\lambda_{\text{accept}} = 1$ so every step falls in the project region, yielding $u_t = (I - P_{\mathcal{S}})g_t$.
- **Hard reject**: $\lambda_{\text{proj}} = 1$ discards all updates ($u_t = 0$).

**With momentum / weight decay.** With momentum $m_t = \beta m_{t-1} + (1 - \beta)u_t$ and weight decay $\lambda$,

$$\theta_{t+1} = (1 - \eta\lambda)\, \theta_t - \eta\, m_t. \tag{10}$$

## 3.2 Monte Carlo Approximation

Computing $\cos\theta$ against all stored gradients is prohibitively expensive when the buffer size $B$ is large. To mitigate this, we maintain a buffer $\{g_i\}_{i=1}^{B}$ of past gradients and randomly sample $k \ll B$ directions at each update:

$$\hat{\cos}\theta = \max_{j=1,\ldots,k} \frac{g_t^\top g_{i_j}}{\|g_t\| \cdot \|g_{i_j}\|}, \quad g_{i_j} \sim \mathcal{S}.$$

This approximation reduces the dot-product complexity from $O(Bd)$ to $O(kd)$ per step, offering a substantial computational savings. Importantly, the sampled maximum is a *conservative* estimate: because only $k$ candidates are considered, $\hat{\cos}\theta$ tends to underestimate the true maximum alignment. While downward-biased in expectation, this bias is benign and even advantageous in practice, as it favors projection or rejection over direct acceptance. Empirically, this conservative tendency aligns with the observed stability gains of our method, providing both efficiency and robustness at no additional cost.

## 3.3 Suppressing Gradient Interference with Selective Projection

Building on Section 2.2, recall that interference occurs when $g_i^\top u < 0$ for a past gradient $g_i$. GEM (Lopez-Paz & Ranzato, 2022) prevents such interference by solving a quadratic program with *inequality constraints* $g^\top u \geq 0$ for stored directions (Eq. 4), projecting $g_t$ onto the corresponding feasible cone. By contrast, OGD (Farajtabar et al., 2019) and GPM (Saha et al., 2021) adopt the stricter *equality-constrained* view, removing all components in the stored subspace $\mathcal{S} = \mathrm{span}(\mathcal{B})$ via the orthogonal update $u = (I - P_\mathcal{S})g_t$ (Eq. 6), which minimizes first-order forgetting for tasks whose gradients lie in $\mathcal{S}$.

SFAO extends these ideas by introducing a *similarity-gated rule* that selects among accept, project, and discard operations. To analyze its guarantees, define the sampled interference risk

$$\widehat{\mathcal{R}}(u;\mathcal{C}) = \max_{g \in \mathcal{C}}(-g^\top u)_+,$$

for a subset $\mathcal{C} \subseteq \mathcal{B}$ of stored directions.

**Project region.** If $u = (I - P_\mathcal{S})g_t$, then $g^\top u = 0$ for all $g \in \mathcal{B}$, hence $\widehat{\mathcal{R}}(u;\mathcal{C}) = 0$. This recovers the first-order safety guarantees of OGD/GPM for tasks represented in $\mathcal{S}$.

**Accept region.** If $\hat{s}_t > \lambda_{\mathrm{accept}} \geq 0$, then even the worst sampled cosine similarity is nonnegative. For the sampled $g^\star$ attaining $\hat{s}_t$ we have $(g^\star)^\top g_t \geq 0$, so $\widehat{\mathcal{R}}(g_t;\mathcal{C}) = 0$. (The restriction $\lambda_{\mathrm{accept}} \geq 0$ is essential; otherwise negative-alignment directions could still be accepted.)

**Discard region.** If $u = 0$, the update is null and trivially safe.

**Conservativeness under sampling.** Since $\hat{s}_t = \max_{g \in \mathcal{C}} \cos(g_t, g) \leq s_t^\star = \max_{g \in \mathcal{B}} \cos(g_t, g)$, sub-sampling provides a deterministic lower bound on the true maximum alignment. Therefore, relative to full-buffer decisions, SFAO with finite $k$ can only *increase* the likelihood of projection or discarding (never reduce it), making the method conservative in suppressing interference.

**Discard region.** $u = 0$ is trivially safe.

Since $\hat{s}_t \leq s_t^\star$, sub-sampling is conservative: relative to decisions made with the full buffer, it can only *increase* the likelihood of projecting or discarding (never reduce it), which further suppresses interference at fixed thresholds.

## 4 Experiments and Results

We evaluate on standard CL benchmarks for comparability with prior work: Split MNIST and Permuted MNIST (LeCun & Cortes, 2005; Goodfellow et al., 2013), Split CIFAR-10/100 (Krizhevsky et al., 2009), and Tiny ImageNet.

**Baselines. (1) OGD** (Farajtabar et al., 2019): A gradient projection method that enforces orthogonality to previously learned parameter subspaces. It is our primary baseline given its geometric alignment with SFAO's projection-based approach. **(2) EWC** (Kirkpatrick et al., 2017): A seminal regularization-based method that constrains parameter updates according to their estimated importance to prior tasks via the Fisher Information Matrix. This provides a representative benchmark for weight-consolidation approaches. **(3) SI** (Zenke et al., 2017): An efficient path-regularization method that computes parameter importance online and penalizes changes to parameters deemed critical for previous tasks. **(4) SGD**: Vanilla stochastic gradient descent, which lacks any mechanism to mitigate catastrophic forgetting, is included as a naive baseline to illustrate the magnitude of improvement achieved by SFAO.

## 4.1 METHOD STABILITY AND ARCHITECTURAL REQUIREMENTS

**Observation.** During initial experiments, we discovered that regularization-based methods EWC and SI exhibited significant instability when paired with lightweight architectures, often diverging or producing invalid losses on the Simple CNN backbone. This instability required switching to more complex architectures to achieve stable training.

**Fix.** We address this by conducting experiments on both architectural settings. Initially, we evaluate geometry-aware methods (OGD and SFAO) on Simple CNN and regularization methods (EWC and SI) on Wide ResNet-28×10 (WRN28×10) due to stability constraints. Subsequently, when computational resources became available, we conducted additional experiments evaluating all methods on WRN28×10 to enable direct comparisons.

**Implication.** While architectural adjustments can resolve stability issues, this approach highlights a fundamental limitation: methods that require specific architectural choices to function properly lack the generalizability needed for real-world deployment. In practice, practitioners cannot always guarantee access to large or specially designed models, making architecture-agnostic stability crucial for continual learning methods.

**New Model Results.** We present results for CIFAR datasets under both experimental settings. The first set of tables shows results with Simple CNN for geometry-aware methods and WRN28×10 for regularization methods. The second set of tables shows all methods evaluated on WRN28×10, enabling direct head-to-head comparisons. SFAO demonstrates consistent performance across both architectural settings without requiring backbone-specific adjustments, positioning it as a more generalizable solution that maintains stability regardless of model capacity constraints.

**Setup.** For MNIST datasets, all baselines use a Simple MLP consisting of a flattened input layer, a single hidden layer with 784 units and ReLU activation, followed by a linear classifier to C classes.

For CIFAR experiments, we present results under two architectural settings. In the first setting, geometry-aware methods (OGD, SFAO, SGD) use a Simple CNN consisting of two convolutional blocks with 3×3 kernels (32 and 64 channels respectively), each followed by ReLU activation and 2×2 max pooling, then a 128-unit fully connected layer and a linear classifier. Regularization methods (EWC, SI) use WRN28×10 with standard formulation including 28 layers, widening factor 10, batch normalization, and residual connections. In the second setting, all methods are evaluated on WRN28×10 to enable direct head-to-head comparisons.

All reported results include standard deviations computed over 5 runs with different random seeds, ensuring statistical reliability while remaining within our compute budget.

**Architectures.** For MNIST datasets, all baselines use a Simple MLP: flattened input $\rightarrow$ a single hidden layer (784 units, ReLU) $\rightarrow$ linear classifier to $C$ classes. For Group (A) CIFAR experiments (OGD, SFAO, SGD) we use a **Simple CNN** consisting of two convolutional blocks with $3 \times 3$ kernels (32 and 64 channels), each followed by ReLU and $2 \times 2$ max pooling, then a 128-unit fully connected layer and a linear classifier. For Group (B) CIFAR experiments (EWC, SI) we use a **WRN28×10** (standard formulation with 28 layers, widening factor 10, batch normalization, and residual connections), which provides the capacity and stability required by these regularization-based methods.

**Hyperparameters.** Across all datasets, we use an SGD optimizer with a momentum of 0.9, a learning rate of $10^{-3}$, batch size of 32, and 2 epochs per task to control compute and isolate forgetting behavior. For EWC and SI, we follow Avalanche's implementation[1] and select regularization strength $\lambda$ by a small grid search on early tasks. For SFAO, we sweep cosine thresholds $\lambda_{\text{proj}}$ and $\lambda_{\text{accept}}$ in the range 0.80–0.95 (discard threshold fixed at $-1 \times 10^{-4}$, max storage capped at 200), and display the best result.

**Compute Efficiency.** All experiments were run on a single NVIDIA A40 GPU (9 vCPUs, 48GB host memory). SFAO introduces minimal overhead—training time increased by less than 6-8% compared to vanilla SGD.

## 4.2 SPLIT MNIST BENCHMARK

| | Accuracy ± Std. Deviation (%) | | | | |
|---|---|---|---|---|---|
| | **Task 1** | **Task 2** | **Task 3** | **Task 4** | **Task 5** |
| SGD | 67.4±0.5 | 75.9±0.8 | 47.4±1.0 | 97.0±0.2 | 91.0±0.3 |
| EWC | 12.8±0.4 | 11.5±0.9 | 31.8±0.7 | 12.0±0.4 | **99.8±0.1** |
| SI | 93.9±0.3 | **92.6±0.5** | **99.3±0.1** | **99.8±0.4** | 99.2±0.1 |
| OGD | **99.9±0.0** | 68.0±1.2 | 54.6±1.0 | 74.7±0.8 | 42.7±1.5 |
| SFAO | 93.6±0.4 | 79.3±0.9 | 47.2±1.1 | 95.6±0.3 | 86.8±0.5 |

Table 1: *Split MNIST*: The accuracy of the model after sequential training on five tasks. The best continual results are highlighted in **bold**.

As shown in Table 1, SI attains the best overall performance with minimal forgetting. SFAO is not as strong as SI or OGD on this benchmark; however, it substantially improves over EWC and SGD in terms of retention while maintaining high per-task accuracy. These results position SFAO as a memory-efficient, geometry-aware optimizer that compares favorably to regularization baselines on MNIST-scale problems.

## 4.3 PERMUTED MNIST BENCHMARK

| | Accuracy ± Std. Deviation (%) | | |
|---|---|---|---|
| | **Task 1** | **Task 2** | **Task 3** |
| SGD | 75.7±0.6 | 81.7±0.4 | 83.5±0.3 |
| EWC | 73.0±0.5 | 75.6±0.7 | 77.4±0.6 |
| SI | **92.8±0.2** | **95.3±0.1** | **94.9±0.1** |
| OGD | 79.3±0.4 | 79.8±0.3 | 81.3±0.4 |
| SFAO | 76.0±0.6 | 79.3±0.5 | 82.8±0.7 |

Table 2: *Permuted MNIST*: The accuracy of the model after sequential training on three permutations $(p_1, p_2, p_3)$. The best continual results are highlighted in **bold**.

As shown in Table 2, SI achieves the highest accuracy across permutations. However, SFAO produces competitive results and outperforms EWC. SFAO also narrows the average accuracy gap with OGD at higher cosine thresholds (see Appendix A.4)

## 4.4 SPLIT CIFAR-100 BENCHMARK (WITHOUT WRN)

We extended Split CIFAR-100 to 10 tasks following the standard protocol. Table 3 reports per-task accuracies for Group A methods on the Simple CNN; Group B methods are shown for context using a WRN28×10. While SFAO underperforms OGD in final accuracy with the Simple CNN backbone, it is notably more consistent across tasks and outperforms OGD on most tasks until the last. This highlights a trade-off: OGD excels at preserving late-task performance, whereas SFAO provides steadier retention throughout training.

---

[1]We build on the open-source `Avalanche` framework (Carta et al., 2023), available at `https://github.com/ContinualAI/continual-learning-baselines/tree/main`.

| | Accuracy ± Std. Deviation (%) | | | | | | | | | |
|---|---|---|---|---|---|---|---|---|---|---|
| | Task 1 | Task 2 | Task 3 | Task 4 | Task 5 | Task 6 | Task 7 | Task 8 | Task 9 | Task 10 |
| SGD | 10.1±0.3 | 10.1±0.3 | 8.0±0.2 | 9.6±0.2 | 10.4±0.2 | 10.1±0.3 | 10.9±0.3 | 9.0±0.2 | 11.4±0.3 | 12.3±0.3 |
| EWC | **19.4**±0.5 | **18.2**±0.4 | 14.5±0.3 | **24.7**±0.5 | **21.6**±0.4 | 18.7±0.3 | 20.9±0.4 | 15.9±0.3 | 22.0±0.4 | 13.5±0.3 |
| SI | 12.2±0.8 | 14.0±0.7 | **19.1**±0.9 | 14.4±0.6 | 16.9±0.7 | **32.3**±1.6 | 28.4±1.3 | **31.5**±2.0 | **37.8**±2.1 | 43.6±3.5 |
| OGD | 8.5±0.2 | 3.6±0.1 | 8.0±0.2 | 6.4±0.2 | 4.5±0.2 | 8.4±0.3 | 21.3±0.5 | 13.6±0.4 | 15.90±1.3 | **66.0**±2.4 |
| SFAO | 8.9±0.3 | 8.3±0.3 | 9.9±0.2 | 11.2±0.2 | 12.5±0.2 | 11.2±0.5 | 26.7±0.8 | 16.8±2.3 | 21.4±1.3 | 23.6±3.8 |

Table 3: *Split CIFAR-100*: The accuracy of the model after sequential training on all ten tasks. The best continual results are highlighted in **bold**.

| | Accuracy ± Std. Deviation (%) | | | | | | | | | |
|---|---|---|---|---|---|---|---|---|---|---|
| | Task 1 | Task 2 | Task 3 | Task 4 | Task 5 | Task 6 | Task 7 | Task 8 | Task 9 | Task 10 |
| SGD | 8.6±0.5 | 3.9±0.7 | 9.0±0.2 | 7.0±0.4 | 10.2±0.3 | 7.2±0.5 | 18.3±0.3 | 8.7±0.4 | 15.2±0.6 | 46.8±0.2 |
| EWC | **19.4**±0.5 | **18.2**±0.4 | 14.5±0.3 | **24.7**±0.5 | **21.6**±0.4 | 18.7±0.3 | 20.9±0.4 | 15.9±0.3 | 22.0±0.4 | 13.5±0.3 |
| SI | 12.2±0.8 | 14.0±0.7 | **19.1**±0.9 | 14.4±0.6 | 16.9±0.7 | **32.3**±1.6 | **28.4**±1.3 | **31.5**±2.0 | **37.8**±2.1 | 43.6±3.5 |
| OGD | 10.8±0.2 | 2.6±0.3 | 7.2±0.2 | 7.5±0.5 | 7.6±0.4 | 5.6±0.2 | 21.6±0.5 | 14.3±0.3 | 10.8±0.5 | **71.4**±1.1 |
| SFAO | 10.1±0.7 | 4.0±0.5 | 9.4±0.3 | 7.6±0.4 | 5.0±0.4 | 7.4±0.6 | 21.0±0.8 | 17.4±1.8 | 19.0±1.7 | 58.1±4.3 |

Table 4: *Split CIFAR-100 with WRN*: The accuracy of the model after sequential training on all ten tasks. The best continual results are highlighted in **bold**.

## 4.5 SPLIT CIFAR-100 BENCHMARK (WITH WRN)

We extended Split CIFAR-100 to 10 tasks following the standard protocol. Table 4 reports per-task accuracies for all methods using the WRN-28×10 backbone, enabling direct comparison across approaches. SFAO is able to demonstrate more consistent retention across earlier tasks and competitive results on mid-sequence tasks. This contrast highlights a trade-off: OGD preserves strong performance on later tasks, whereas SFAO provides steadier performance throughout training. This indicates SFAO achieves a more balanced performance across the sequence, which may be preferable in applications where uniform retention is important.

## 4.6 SPLIT CIFAR-10 BENCHMARK (WITHOUT WRN)

| | Simple CNN | | | | |
|---|---|---|---|---|---|
| | Task 1 | Task 2 | Task 3 | Task 4 | Task 5 |
| SGD | 49.5±2.3 | 50.0±1.8 | 50.0±2.1 | 50.0±1.5 | 50.0±2.0 |
| EWC | 20.6±1.2 | 17.5±0.9 | 19.2±1.0 | 24.5±1.8 | 23.6±1.1 |
| SI | 70.2±2.7 | 51.8±2.5 | 44.1±2.0 | **66.3**±2.8 | **96.1**±1.5 |
| OGD | **79.3**±3.1 | 58.0±2.7 | 51.6±2.5 | 58.0±3.0 | 93.0±1.2 |
| SFAO | 76.5±2.9 | **62.4**±3.2 | **52.6**±2.4 | 57.6±3.0 | 77.0±2.1 |

| | WRN-28×10 | | | | |
|---|---|---|---|---|---|
| | Task 1 | Task 2 | Task 3 | Task 4 | Task 5 |
| SGD | 77.3±2.3 | 60.4±1.8 | 52.5±2.1 | 51.6±1.5 | 86.3±2.0 |
| EWC | 20.6±1.2 | 17.5±0.9 | 19.2±1.0 | 24.5±1.8 | 23.6±1.1 |
| SI | 70.2±2.7 | 51.8±2.5 | 44.1±2.0 | 66.3±2.8 | **96.1**±1.5 |
| OGD | **80.3**±3.1 | **63.7**±2.7 | 53.0±2.5 | 66.0±3.0 | 94.7±1.2 |
| SFAO | 78.7±2.9 | 56.9±3.2 | **55.4**±2.4 | **69.9**±3.0 | 90.9±2.1 |

Table 5: Split CIFAR-10 benchmark with Simple CNN backbone.

Table 6: Split CIFAR-10 benchmark with WRN-28×10 backbone.

Table 4.6.5 reports per-task accuracies for Group A methods (OGD, SFAO, SGD) evaluated on the Simple CNN; EWC and SI are shown for context using a WRN28×10 and should be treated as qualitative context.[2] Under the lightweight Simple CNN backbone (head-to-head comparison), OGD attains the highest average accuracy overall in our run, while SFAO is competitive on average. This pattern illustrates the stability–plasticity trade-off: OGD can strongly preserve earlier task performance in certain settings, whereas SFAO provides more balanced per-task behavior and reduced projection frequency (see Appendix A.3). We therefore report Group A as direct comparisons and treat Group B as qualitative context only.

## 4.7 SPLIT CIFAR-10 BENCHMARK (WITH WRN)

Table 4.6.6 reports per-task accuracies for all baselines using the WRN-28×10 backbone, enabling direct comparison across methods. SFAO shows strong and balanced performance across the sequence, achieving the best results on mid-sequence tasks (Task 3 and Task 4) and remaining competitive on the first and last tasks. While SI reaches the highest accuracy on the final task, its

---

[2]EWC and SI were evaluated on Wide ResNet-28×10 due to instability / divergence observed on the Simple CNN; see the Setup paragraph.

earlier performance lags behind SFAO. These results highlight that SFAO achieves a favorable balance between stability and plasticity on Split CIFAR-10, outperforming OGD in several tasks while maintaining consistency throughout training.

## 4.8 SPLIT TINYIMAGENET BENCHMARK (WITH WRN)

| | Accuracy ± Std. Deviation (%) | | | | | | | | | |
|---|---|---|---|---|---|---|---|---|---|---|
| | Task 1 | Task 2 | Task 3 | Task 4 | Task 5 | Task 6 | Task 7 | Task 8 | Task 9 | Task 10 |
| SGD | 17.4±1.4 | 19.0±0.7 | 16.3±0.9 | 16.9±0.5 | 19.8±1.0 | 17.3±0.5 | 14.6±1.4 | 18.8±0.4 | 17.3±0.7 | 18.3±1.2 |
| EWC | 23.8±0.0 | 25.0±0.0 | 21.3±0.0 | 18.2±0.0 | 25.7±0.0 | 23.2±0.0 | 19.6±0.0 | 22.9±0.0 | 18.5±0.0 | 22.9±0.0 |
| SI | 5.3±0.0 | 6.1±0.0 | 6.5±0.0 | 8.4±0.0 | 12.1±0.0 | 18.7±0.0 | 18.5±0.0 | **29.1**±0.0 | **35.2**±0.0 | **52.3**±0.0 |
| OGD | 7.5±1.2 | 9.5±1.9 | 10.8±1.4 | 16.2±1.3 | 14.5±2.4 | 20.4±2.8 | 20.7±2.1 | 32.2±3.0 | 31.4±2.2 | 45.5±2.0 |
| SFAO | **24.36**±0.46 | **25.76**±0.81 | **25.30**±1.35 | **24.50**±0.87 | **29.02**±1.60 | **27.54**±1.53 | **25.08**±0.95 | 27.80±1.46 | 26.94±1.13 | 26.30±1.53 |

Table 7: *Split TinyImageNet*: The accuracy of the model after sequential training on all ten tasks.

Table 7 shows that SFAO is competitive on early tasks of Split TinyImageNet, whereas SI excels on the final three tasks and EWC remains strong in the first half. Given the benchmark's greater complexity (fine-grained categories, higher intra-class variation, and stronger distribution shifts), these trends may reflect differing robustness profiles across difficulty regimes rather than a single global ranking. A plausible explanation is that SFAO's accept/project mechanism favors rapid adaptation early in the stream, while regularization-based approaches (SI/EWC) offer greater stability later; a definitive causal analysis is left to future work.

## 5 LIMITATIONS AND FUTURE DIRECTIONS

### 5.1 ARCHITECTURAL GENERALIZABILITY

A key limitation was the instability of regularization-based methods like EWC and SI, requiring us to switch to a WRN28×10 backbone for stable training. This highlights the need for methods robust across diverse architectures and model capacities. While SFAO shows architecture-agnostic stability, the field needs systematic approaches ensuring method robustness without architectural workarounds. Future work should develop continual learning techniques maintaining consistent performance across varying model sizes, enabling deployment in resource-constrained scenarios.

### 5.2 TASK ORDERING EFFECTS

Continual learning performance often depends on task sequence, with some orders amplifying forgetting and others resembling curricula (Bell & Lawrence, 2022; Kemker et al., 2018). Since SFAO regulates updates through thresholds, future work could explore *dynamic robustness* via checkpoints and backtracking: if a new task induces sharp forgetting, training can revert and continue with stricter thresholds, effectively "learning more cautiously." Threshold statistics also provide a proxy for task difficulty, enabling automated adaptation and the design of optimal curricula. Thus, SFAO could both mitigate order sensitivity and serve as a principled tool for quantifying and improving task sequencing across continual learning methods.

### 5.3 PER-LAYER THRESHOLD TRAINING

Beyond fixed thresholds, a promising direction is **learning thresholds dynamically**. Thresholds $\lambda_{\text{proj}}^{\ell}$ and $\lambda_{\text{accept}}^{\ell}$ can be treated as learnable parameters and optimized via backpropagation with differentiable gating (e.g., sigmoid soft thresholds) or via reinforcement learning (Ghasemi & Ebrahimi, 2024) using long-term metrics like forgetting and compute cost.

### 5.4 DYNAMICALLY UPDATE AND SCHEDULE THRESHOLDS

Thresholds can be updated with learning rates or schedules, becoming stricter near convergence to reduce interference and improve stability. Strategies include linear warm-up with exponential growth (Kalra & Barkeshli, 2024) or piecewise updates (Cohen-Addad & Kanade, 2016). Thresholds can also adapt to performance metrics such as forgetting rate or plasticity–stability scores for dynamic sensitivity control.

## 6 RELATED WORK

### 6.1 GEOMETRY-AWARE METHODS

The geometry-aware perspective in continual learning began as an alternative to memory replay and regularization. Instead of storing data or penalizing parameter shifts, methods like OGD proposed projecting gradients onto subspaces orthogonal to prior tasks, ensuring updates do not interfere with previous knowledge (Farajtabar et al., 2019). This concept was further refined by Gradient Projection Memory (GPM), which used Singular Value Decomposition (SVD) to build compact gradient subspaces and selectively project future updates (Cha et al., 2020). These methods often rely on operations such as orthogonalization or SVD. Although effective, such approaches introduce structural overhead that SFAO addresses through lightweight probabilistic approximations of gradient alignment.

### 6.2 REGULARIZATION-BASED METHODS

Regularization-based methods such as EWC and SI were among the first to gain traction to address catastrophic forgetting (Kirkpatrick et al., 2017; Zenke et al., 2017). They constrain updates to important parameters using gradient tracking metrics by imposing static penalties (e.g., quadratic loss terms) based on parameter sensitivity. Some recent variants, such as RTRA, combine regularization with adaptive gradient strategies to improve stability and training efficiency (Zhao et al., 2023). These methods model forgetting as a function of parameter importance, introducing fixed or adaptive constraints during optimization. Our work differs in that SFAO modulates updates dynamically based on local alignment with previously learned gradient directions.

### 6.3 THEORETICAL PERSPECTIVES ON FORGETTING

A growing body of work aims to dissect why catastrophic forgetting occurs in neural networks. Early empirical studies suggest that standard gradient descent optimizers completely overwrite earlier task knowledge (Goodfellow et al., 2013). Later papers like (Nguyen et al., 2019) and (Wu et al., 2024) show that forgetting also correlates with gradient interference, task similarity, and network capacity. Our method is grounded in this insight, as SFAO addresses the most cited cause of forgetting, gradient interference by filtering out the conflicting directions during learning. Its cosine similarity testing and projection filtering mechanism are rooted in the theoretical observation that overlapping gradients lead to interference.

## 7 CONCLUSION

We introduce SFAO, a tunable, similarity-gated extension to OGD that balances forgetting and adaptability using cosine similarity. It employs a practical gating mechanism with interpretable parameters to regulate stability, ensuring consistent memory retention under a fixed compute budget. This design also provides a promising path toward adaptive or scheduled thresholds, offering flexible control strategies in continual learning. SFAO integrates seamlessly with SGD, without requiring additional losses, memory buffers, or architectural overhead.

## 8 IMPACT STATEMENT

This work aims to advance the field of machine learning through methodological contributions. We do not identify specific societal or ethical risks arising from this study beyond those typical of general machine learning research.

## 9 REPRODUCIBILITY STATEMENT

All experimental code, hyperparameters, and model configurations are provided to ensure reproducibility, and can be found publicly on GitHub at `https://anonymous.4open.science/r/sfao-4E83/`.

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

# A ADDITIONAL EXPERIMENTS

## A.1 FORGETTING ON SPLIT MNIST

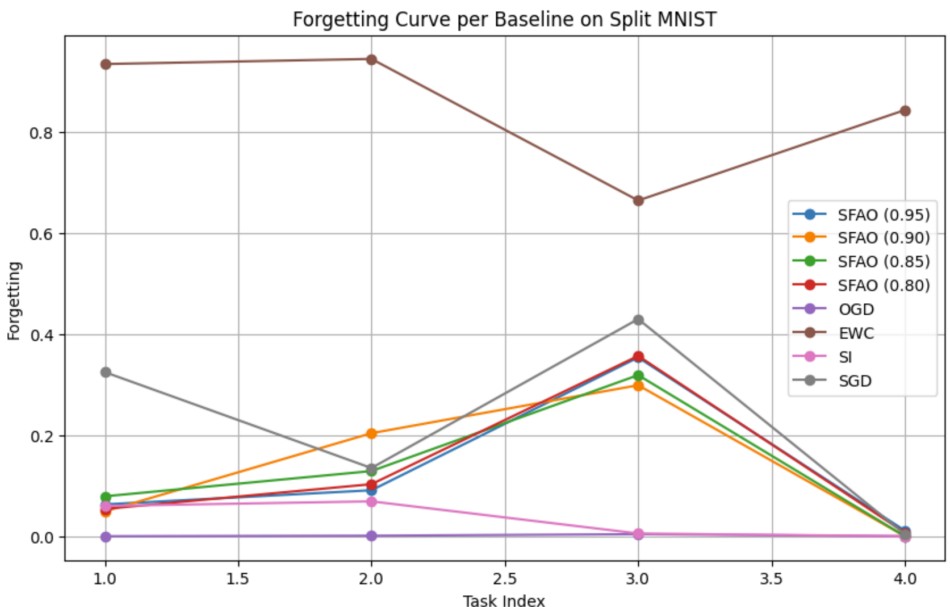

Figure 1: Forgetting curve per baseline on Split MNIST. Forgetting is averaged across previously seen tasks after each new task. There are a total of four tasks.

## A.2 SFAO AND OGD MEMORY USAGE COMPARISON

The memory usage was calculated using in the form of megabytes (MB):

$$\text{Memory (MB)} = \frac{|\mathcal{S}| \times \text{num\_params} \times 4}{1024^2}$$

where $|\mathcal{S}|$ is the number of stored gradients, num_params is the total number of model parameters, and $4$ is the number of bytes per `float32`.

| Dataset | OGD (MB) | SFAO (MB) |
|---|---|---|
| Split MNIST | 1441.82 | **153.71** |
| Permuted MNIST (3) | 4367.28 | **155.28** |
| Permuted MNIST (5) | 7278.00 | **155.28** |

Table 8: Memory usage (MB) comparison between OGD and SFAO across Split MNIST and Permuted MNIST. For Permuted MNIST, experiments were conducted with $p_1$–$p_3$ permutations (3) and $p_1$–$p_5$ permutations (5)

As seen in Table 8, SFAO substantially reduces memory usage on Split MNIST and Permuted MNIST, remaining essentially constant across increasing permutations. This efficiency stems from SFAO's buffer management strategy: the cosine similarity threshold prevents redundant gradients from entering the buffer, while the discard threshold removes uninformative vectors, keeping $|\mathcal{S}|$ bounded regardless of the number of tasks. On Split CIFAR-100, SFAO uses slightly more memory than OGD due to higher-dimensional and more diverse gradients, which fewer pass the filtering thresholds. This modest increase reflects a trade-off that prioritizes stability and mitigates catastrophic forgetting in complex datasets, demonstrating that SFAO balances efficiency and reliability across different benchmarks.

| Dataset | OGD | SFAO |
|---|---|---|
| Split MNIST | 5625 | 200 |
| Permuted MNIST | 5625 | 200 |
| Split CIFAR-100 | 300* | 200 |

Table 9: Projection frequency per batch for OGD and SFAO across benchmarks. *For Split CIFAR-100, OGD uses a capped gradient memory (`max_mem_dirs` = 1000) and harvest policy (`dirs_per_task` = 120, `harvest_batches` = 30), unlike MNIST where projections scale with the full stored gradient set.

## A.3 Average Projection Frequency

As seen in Table 9 We observe that OGD incurs significantly higher projection counts, especially on MNIST benchmarks where projections scale with the full memory of past gradients. In contrast, SFAO maintains a fixed low projection frequency across all tasks, offering a more computationally efficient alternative. While OGD's capped memory reduces this burden on Split CIFAR-100, SFAO still provides stable performance with substantially fewer projections.

## A.4 Different Cosine Similarity Thresholds vs OGD Accuracy

| Dataset | OGD | SFAO (0.95) | SFAO (0.90) | SFAO (0.85) | SFAO (0.80) |
|---|---|---|---|---|---|
| Permuted MNIST (3) | 0.8014 | 0.7815 | 0.7753 | 0.7938 | 0.7815 |
| Permuted MNIST (5) | 0.7933 | 0.7633 | 0.7612 | 0.7799 | 0.7887 |
| Split CIFAR-10 | 0.6800 | 0.6525 | 0.6487 | 0.6152 | 0.6219 |
| Split CIFAR-100 | 0.1562 | 0.1368 | 0.1500 | 0.1436 | 0.1505 |

Table 10: Average accuracy comparison of OGD and SFAO across different cosine similarity thresholds on multiple benchmarks. For Permuted MNIST, experiments were conducted with $p_1$–$p_3$ (3 permutations) and $p_1$–$p_5$ (5 permutations).

As seen in Table 10, SFAO demonstrates competitive performance across most datasets, particularly for Permuted MNIST, where thresholds of 0.85 and 0.80 remain close to OGD despite the increased complexity from additional permutations. While OGD generally outperforms SFAO on CIFAR-based benchmarks, the gap is minimal for Split CIFAR-10 and narrows further at lower thresholds (0.80). These results highlight that adaptive cosine thresholds help maintain stability without significantly compromising accuracy, even under more challenging task permutations.

## A.5 Plasticity-Stability Measure

The Plasticity-Stability Measure (PSM) is a scalar metric that quantifies the trade-off between a model's ability to acquire new knowledge (plasticity) and its ability to retain previously learned knowledge (stability). Formally, it is defined as:

$$\text{PSM} = \frac{A_{\text{final}} + A_{\text{avg}}}{2},$$

where $A_{\text{final}}$ is the final accuracy on the last task and $A_{\text{avg}}$ is the average accuracy across all tasks. Higher values indicate a better balance, while lower values suggest excessive forgetting or limited adaptability.

As seen in Table 11, SFAO consistently achieves mid-range PSM values across all benchmarks, remaining close to the balance point between 0 and 1. This reflects its design choice of prioritizing stability while still maintaining sufficient plasticity to adapt to new tasks. However, OGD's behavior varies: on MNIST-scale datasets it favors plasticity, while on high-dimensional datasets like CIFAR it skews heavily toward stability at the cost of adaptability. Overall, SFAO's selective gating yields a steadier stability–plasticity trade-off, making it more reliable across diverse benchmarks.

| Dataset | OGD | SFAO (0.95) | SFAO (0.9) | SFAO (0.85) | SFAO (0.8) |
|---|---|---|---|---|---|
| Split MNIST | 0.4995 | 0.4352 | 0.4310 | 0.4344 | 0.4350 |
| Permuted MNIST (3) | 0.4999 | 0.4783 | 0.4786 | 0.4897 | 0.4791 |
| Permuted MNIST (5) | 0.4958 | 0.4683 | 0.4592 | 0.4742 | 0.4769 |
| CIFAR-100 | 0.2511 | 0.4691 | 0.4636 | 0.4768 | 0.4671 |
| CIFAR-10 | 0.3574 | 0.4593 | 0.4454 | 0.4277 | 0.4320 |

Table 11: Plasticity-Stability Comparison of OGD and SFAO across different cosine similarity thresholds on multiple benchmarks. For Permuted MNIST, experiments were conducted with $p_1$–$p_3$ (3 permutations) and $p_1$–$p_5$ (5 permutations).

# B  ALGORITHMS

## B.1  SFAO (SIMILARITY-GATED UPDATE WITH MONTE CARLO SAMPLING)

---

**Algorithm 1** SFAO: Single-layer similarity-gated update (per step)

---

**Require:** Current gradient $g_t \in \mathbb{R}^d$; buffer $\mathcal{B} = \{g_i\}_{i=1}^B$; thresholds $\lambda_{\mathrm{proj}} \leq \lambda_{\mathrm{accept}}$; Monte Carlo sample size $k \ll B$; buffer policy parameters $(B_{\max}, \tau_{\mathrm{add}}, \tau_{\mathrm{drop}})$

**Ensure:** Update direction $u_t$ and updated buffer $\mathcal{B}$

  0: $\mathcal{C} \leftarrow$ SAMPLESUBSET$(\mathcal{B}, k)$ {uniform without replacement}

  0: $\hat{s} \leftarrow$ MCMAXCOS$(g_t, \mathcal{C})$ {$\hat{s} = \max_{g \in \mathcal{C}} \frac{g_t^\top g}{\|g_t\|\|g\|}$ (conservative)}

  0: **if** $\hat{s} > \lambda_{\mathrm{accept}}$ **then** {accept}

  0:   $u_t \leftarrow g_t$

  0: **else if** $\lambda_{\mathrm{proj}} < \hat{s} \leq \lambda_{\mathrm{accept}}$ **then** {project}

  0:   $u_t \leftarrow (I - P_{\mathcal{S}})\, g_t$ {$\mathcal{S} = \mathrm{span}(\mathcal{B})$}

  0: **else**{discard}

  0:   $u_t \leftarrow 0$

  0: **end if**

  0: $\mathcal{B} \leftarrow$ UPDATEBUFFER$(\mathcal{B}, g_t, B_{\max}, \tau_{\mathrm{add}}, \tau_{\mathrm{drop}})$

  0: **return** $u_t, \mathcal{B}$ =0

---

## B.2  GEOMETRY OF THE SFAO UPDATE

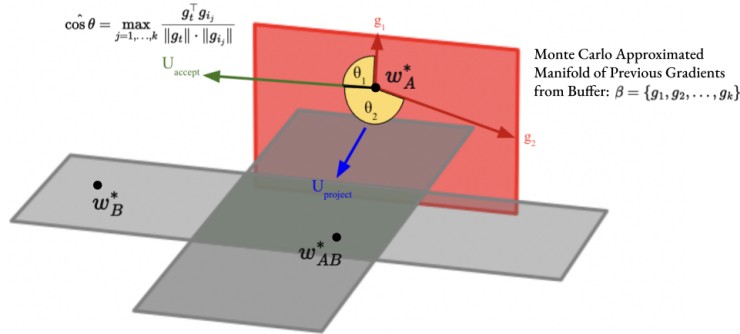

Figure 2: Geometry of the SFAO update. Green ($U_{\mathrm{accept}}$): when the current gradient is sufficiently similar to the buffer $\mathcal{B}$, the update is accepted as is. Blue ($U_{\mathrm{project}}$): otherwise the gradient is orthogonally projected off the subspace spanned by the buffered past gradients $\{g_i\}$ to mitigate interference.

### B.3 PER-LAYER SFAO: MATHEMATICAL FORMULATION AND ALGORITHM

**Mathematical formulation.** For layer $\ell \in \{1, \ldots, L\}$, let $g_t^{(\ell)}$ be the layer-wise gradient and $\mathcal{B}^{(\ell)} \subset \mathbb{R}^{d_\ell}$ its buffer. With Monte Carlo subset $\mathcal{C}^{(\ell)} \subset \mathcal{B}^{(\ell)}$ of size $k_\ell$, define

$$s^{(\ell)} = \max_{g \in \mathcal{C}^{(\ell)}} \frac{\langle g_t^{(\ell)}, g \rangle}{\|g_t^{(\ell)}\| \, \|g\|}.$$

Given thresholds $-1 \leq \lambda_{\text{proj}}^{(\ell)} \leq \lambda_{\text{accept}}^{(\ell)} \leq 1$, set the layer update

$$\mathcal{U}^{(\ell)}\left(g_t^{(\ell)}\right) = \begin{cases} g_t^{(\ell)}, & s^{(\ell)} > \lambda_{\text{accept}}^{(\ell)} \\ \left(I - P_{\mathcal{S}^{(\ell)}}\right) g_t^{(\ell)}, & \lambda_{\text{proj}}^{(\ell)} < s^{(\ell)} \leq \lambda_{\text{accept}}^{(\ell)} \\ 0, & s^{(\ell)} \leq \lambda_{\text{proj}}^{(\ell)} \end{cases} \quad \text{with } \mathcal{S}^{(\ell)} = \text{span}\big(\mathcal{B}^{(\ell)}\big).$$

Concatenate (or assemble) per-layer updates to obtain $u_t = \big(\mathcal{U}^{(1)}(g_t^{(1)}), \ldots, \mathcal{U}^{(L)}(g_t^{(L)})\big)$ and update parameters $\theta \leftarrow \theta - \eta \, u_t$ per SGD.

## C ADDITIONAL RESULTS AND PROOFS

### C.1 MINIMIZING GRADIENT INTERFERENCE RISK

Recall Eq. 5 for minimizing the *interference risk* of an update $u$ against a set $G \subset \mathbb{R}^d$ of stored directions. Here, we solve the constrained optimization problem

$$\min_{u \in \mathbb{R}^d} \frac{1}{2}\|u - g_t\|_2^2 \quad \text{s.t.} \quad g^\top u = 0 \quad \forall g \in G,$$

We proceed by solving the Lagrangian under the formal constraint $G^\top u = 0$:

$$\mathcal{L}(u, \lambda) = \frac{1}{2}\|u - g_t\|_2^2 + \lambda^\top (G^\top u) \tag{11}$$

Next, we evaluate the Karush–Kuhn–Tucker (KKT) conditions:
**Stationarity:**

$$\nabla_u \mathcal{L}(u^*, \lambda^*) = \nabla_u \left( \frac{1}{2}\|u - g_t\|_2^2 + \lambda^\top (G^\top u) \right) = 0 \tag{12}$$

$$= u - g_t + G\lambda = 0 \tag{13}$$

$$\implies u^* = g_t - G\lambda \tag{14}$$

**Primal Feasibility:**

$$G^\top u = 0 \tag{15}$$

$$G^\top (g_t - G\lambda) = 0 \quad \text{per Stationarity} \tag{16}$$

$$G^\top g_t - G^\top G\lambda = 0 \tag{17}$$

$$G^\top g_t = G^\top G\lambda \tag{18}$$

$$\implies \lambda^* = (G^\top G)^\dagger G^\top g_t \tag{19}$$

Since our problem only involves linear equality constraints, the multipliers $\lambda$ are unconstrained and all equalities are always active, so the dual feasibility and complementary slackness conditions are vacuous and need not be checked. Also, note that $\dagger$ denotes the Moore-Penrose Pseudoinverse.

Substituting $\lambda^*$:

$$u^* = g_t - G(G^\top G)^\dagger G^\top g_t \tag{20}$$

$$\implies u^* = (I - G(G^\top G)^\dagger G^\top)g_t \tag{21}$$

Letting $P_{\mathcal{S}} = G(G^\top G)^\dagger G^\top$, we recover Eq. 6:

$$u^* = (I - P_{\mathcal{S}})g_t,$$

which shows that the optimal update is the projection of the current gradient step $g_t$ onto the orthogonal complement of the span of past gradients.

**SVD expression.**   Let the thin SVD of $G \in \mathbb{R}^{d \times k}$ be

$$G = U_r \Sigma_r V_r^\top,$$

where $r = \mathrm{rank}(G)$, $U_r \in \mathbb{R}^{d \times r}$ and $V_r \in \mathbb{R}^{k \times r}$ have orthonormal columns, and $\Sigma_r \in \mathbb{R}^{r \times r}$ is diagonal with positive entries. Then

$$G^\top G = V_r \Sigma_r^2 V_r^\top \quad \Rightarrow \quad (G^\top G)^\dagger = V_r \Sigma_r^{-2} V_r^\top,$$

and hence

$$P_\mathcal{S} = G(G^\top G)^\dagger G^\top = (U_r \Sigma_r V_r^\top)(V_r \Sigma_r^{-2} V_r^\top)(V_r \Sigma_r U_r^\top) = U_r U_r^\top.$$

Therefore, the optimal update can be written purely in terms of the left singular vectors of $G$:

$$\boxed{u^\star = (I - U_r U_r^\top)\, g_t.}$$

