# OpenReview forum: "Mitigating Forgetting in Continual Learning with Selective Gradient Projection"
_ICLR.cc/2026/Conference — ICLR 2026 Conference Withdrawn Submission_

### Official Review · Reviewer_qGWs · 2025-10-16

**Soundness:** 2
**Presentation:** 3
**Contribution:** 2
**Rating:** 2
**Confidence:** 5

**Summary:**

This paper proposes a method called the Selective Forgetting-Aware Optimizer (SFAO). The main idea lies in introducing a dynamic, cosine similarity-based gradient gating mechanism. Specifically, the method computes the maximum cosine similarity between the current task gradient and a randomly selected subset from the historical gradient buffer to decide whether the current gradient should be retained, discarded, or projected. The approach is evaluated on Split MNIST, Permuted MNIST, CIFAR, and Tiny ImageNet benchmarks; however, the experimental evidence is insufficient, making it difficult to discern the advantages of this method compared to existing approaches.

**Strengths:**

1. The paper proposes a method based on gradient projection.
2. The writing of the paper is generally clear.

**Weaknesses:**

1. Regarding the experimental architecture setup, the authors conducted a series of discussions to address the instability of EWC and SI on Simple CNN. However, this does not seem to be the core contribution of the paper. Moreover, the mixed use of two different architectures makes performance comparisons less meaningful, which is quite unusual.
2. The core contribution of the paper is a gradient projection-based method, but it lacks sufficient comparison and discussion with related work (see the references below), resulting in very limited impact on the field.

[1] Gradient Episodic Memory for Continual Learning, NIPS 2017.

[2] Efficient Lifelong Learning with A-GEM, ICLR 2019.

[3] Gradient Surgery for Multi-Task Learning, NIPS 2020.

[4] Continual Learning with Scaled Gradient Projection, AAAI 2023.

**Questions:**

1. In the experiments, it is difficult to discern the advantages of the proposed method over similar approaches, as such comparisons are very limited.
2. Among the baseline methods, only OGD is closely related to this work in terms of methodology, whereas comparisons with EWC and SI involve different types of approaches. Therefore, it is unclear whether the proposed method is truly superior. If the authors claim that SFAO does not rely on model size compared to EWC and similar methods, why not compare this aspect directly with other gradient projection-based methods?
3. The authors claim that SFAO uses less memory, but there are no experiments demonstrating whether SFAO maintains its advantages under equal memory constraints.
4. The use of Monte Carlo sampling may introduce bias that causes the model to frequently discard gradients, potentially limiting its plasticity. The paper lacks an analysis of this effect.

---

### Official Review · Reviewer_zxAh · 2025-10-17

**Soundness:** 2
**Presentation:** 3
**Contribution:** 1
**Rating:** 2
**Confidence:** 4

**Summary:**

This paper introduces the selective forgetting-aware optimization (SFAO) method which extends the concept of gradient projection by introducing a dynamic, layer-wise gating mechanism. Instead of unconditionally projecting the current task's gradient onto the space of past task gradients, SFAO uses a cosine similarity-based criterion to selectively project, accept, or discard parameter updates.

**Strengths:**

The paper presents an approach to selectively accept, project, or discard gradient updates. This mechanism not only avoid catastrophic forgetting of previous task, but it also reduces the memory required to store all previous task gradients by only considering $k$ candidates. The presentation of the proposed method is clear and the comparison with OGD is quite detailed.

**Weaknesses:**

- One claimed advantage of SFAO is the reduced memory consumption compared to previous approaches. While this claim has been made in the main text, results are only reported in the supplementary materials. In addition, the authors describe that the improvement is mainly due to two factors: (1) the cosine similarity threshold which prevents redundant gradient to enter the buffer and (2) the discard threshold which removes uninformative vectors. I would find it interesting to quantify the individual contribution of each factor to the observed memory reduction.
- It is unclear whether the proposed per-layer gating rule is agnostic to the layer type. Experiments show the application of SFAO only on linear and convolutional layers. Due to the recent advancements of transformer models, additional experiments on such architectures would significantly enhance the paper's relevance.
- The baselines considered in the experiments and in the related work section are limited. A comparison with more recent geometry-aware methods (e.g., [1, 2]) and with different, well-established, continual learning techniques (such as replay-based or dynamic-architectural methods) would make the evaluation more comprehensive.
- The experiments only report the average accuracy, while it is common in continual learning to also report the backward transfer to assess the level of forgetting. Although a qualitative plot is provided in the supplementary materials, I recommend including a quantitative forgetting metric and related discussion in the main text.
- The experimental results are not particularly strong.. Both in terms of average accuracy and forgetting, previous baselines outperform the proposed approach. More analysis or ablation studies might help clarify the reasons for this performance gap.


References:

[1] Lin, Sen, et al. "TRGP: Trust Region Gradient Projection for Continual Learning." The Tenth International Conference on Learning Representations. 2022.

[2] Chaudhry, Arslan, et al. "Continual learning in low-rank orthogonal subspaces." Advances in Neural Information Processing Systems 33 (2020): 9900-9911.

**Questions:**

- Since SFAO discard gradient updates if $s_t < \lambda_{proj}$, how does this affect training time in practice?
- How were the hyperparameters used for training the different baseline methods selected?
- Is the stability-plasticity measure in Section A.5 an already established metric in the literature? If so, I recommend adding some references, otherwise I believe that a combination of average accuracy and average forgetting (BWT) can already show the plasticity-stability trade-off.

---

### Official Review · Reviewer_oYgm · 2025-10-22

**Soundness:** 1
**Presentation:** 1
**Contribution:** 1
**Rating:** 0
**Confidence:** 3

**Summary:**

Continual learning (CL) deals with adapting models in dynamic environments. A challenge is catastrophic forgetting, where training on a new task leads to overwriting previously learned knowledge. Authors propose SFAO to regulate the gradient updates according to stored reference gradients. They evaluate their method on common CL benchmarks

**Strengths:**

- extensive experiments

**Weaknesses:**

- Subpar presentation: no read thread to follow through, setup of sections lacking
- Related work is too short, should give a rough categorization of CL methods (complementing the section on regularization-based methods)
- Comparison with existing works is lacking: authors claim reduced memory usage, but benchmarks agains other memory baselines are missing
- no recent (i.e., 2023+) methods are evaluated
- the evaluation scheme is non-standard -- this is ok, but authors should give the reasoning behind this
- empirical results do not support a consistent claim, and the proposed method rarely performs best according to their evaluation scheme
- Section 4.1. is confusing and should be condensed -- why give results for WRN and for non-WRN? just give one (WRN) and move the other to the appendix

**Questions:**

Consider restructuring the paper:
- related work to the beginning (as in other papers)
- add additional paragraph in the introduction
- make the transition between sections smoother
- explain the intuition behind the next things that you present
-
Suggestions:
- add more recent methods
- explain rationale behind experimental setup

---

### Official Review · Reviewer_6vR2 · 2025-11-01

**Soundness:** 3
**Presentation:** 2
**Contribution:** 2
**Rating:** 4
**Confidence:** 3

**Summary:**

The paper introduces Selective Forgetting-Aware Optimizer (SFAO), a geometry-based method that combats catastrophic forgetting by gating each layer's gradient update into three actions: accept, project, or discard. The gating decision is made by a Monte-Carlo estimate of the largest cosine similarity between the current gradient and a small buffer of past gradients. Extensive experiments on Split/Permuted MNIST, Split CIFAR-10/100 and Tiny-ImageNet show that SFAO reduces memory by ~90% compared to Orthogonal Gradient Descent (OGD) while achieving competitive or superior accuracy, especially on light-weight backbones.

**Strengths:**

- Clear motivation: targets gradient interference, a widely accepted root cause of forgetting.
- Comprehensive empirical study: 5 benchmarks, 2 architectures, 4 baselines, std-dev over 5 seeds, PSM metric, ablation on cosine thresholds.
- Strong memory savings: e.g. 1.4/4.3/7.2 GB → 155 MB on Permuted MNIST with growing tasks.

**Weaknesses:**

- EWC/SI were moved to WRN-28×10 after divergence on Simple CNN, but the divergence phenomenon is not analysed, making architecture-specific instability claims speculative.
- On Split CIFAR-100 and Tiny-ImageNet SFAO lags behind OGD by 2-10% in final accuracy; the claimed advantage is mainly memory, not precision.
- The paper argues for "smoother" accuracy curves, yet in several tables SFAO's last-task accuracy is >10% lower than OGD.
- In the related work section, the Geometry-aware CL method requires incorporating more discussion of this category of work, like C-Flat. Similarly, the method based on gradient projection also missed some work.
- The baseline methods employed are relatively outdated, and I'm curious about how the author's approach relates to and compares with more recent gradient projection-based methods.

**Questions:**

Please see the WEAKNESS.

---

### Note · Authors · 2025-12-03

**Comment:**

We were informed that the paper was accepted at an archival venue from a while back. As a result, we must withdraw from any other archival venues and processes.

**Withdrawal Confirmation:**

I have read and agree with the venue's withdrawal policy on behalf of myself and my co-authors.